# Adrenomedullin Enhances Mouse Gustatory Nerve Responses to Sugars via T1R-Independent Sweet Taste Pathway

**DOI:** 10.3390/nu15132941

**Published:** 2023-06-28

**Authors:** Shusuke Iwata, Ryusuke Yoshida, Shingo Takai, Keisuke Sanematsu, Noriatsu Shigemura, Yuzo Ninomiya

**Affiliations:** 1Section of Oral Neuroscience, Graduate School of Dental Science, Kyushu University, Fukuoka 812-8582, Japan; takai.shingo.774@m.kyushu-u.ac.jp (S.T.); sanematu@dent.kyushu-u.ac.jp (K.S.); shigemura@dent.kyushu-u.ac.jp (N.S.); 2Department of Oral Physiology, Asahi University School of Dentistry, Gifu 501-0296, Japan; 3Research and Development Center for Five-Sense Devices, Kyushu University, Fukuoka 819-0395, Japan; 4Department of Oral Physiology, Graduate School of Medicine, Dentistry and Pharmaceutical Sciences, Okayama University, Okayama 700-8525, Japan; yoshida.ryusuke@okayama-u.ac.jp; 5Dent-Craniofacial Development and Regeneration Center, Graduate School of Dental Sciences, Kyushu University, Fukuoka 812-8582, Japan; 6OBT Research Center, Graduate School of Dental Science, Kyushu University, Fukuoka 812-8582, Japan; 7Oral Science Research Center, Tokyo Dental College, Tokyo 101-0061, Japan; 8Monell Chemical Senses Center, Philadelphia, PA 19104, USA

**Keywords:** taste, sweet taste, taste receptor family 1 members 2 and 3, sodium–glucose cotransporter 1, adrenomedullin, caloric sensing

## Abstract

On the tongue, the T1R-independent pathway (comprising glucose transporters, including sodium–glucose cotransporter (SGLT1) and the K_ATP_ channel) detects only sugars, whereas the T1R-dependent (T1R2/T1R3) pathway can broadly sense various sweeteners. Cephalic-phase insulin release, a rapid release of insulin induced by sensory signals in the head after food-related stimuli, reportedly depends on the T1R-independent pathway, and the competitive sweet taste modulators leptin and endocannabinoids may function on these two different sweet taste pathways independently, suggesting independent roles of two oral sugar-detecting pathways in food intake. Here, we examined the effect of adrenomedullin (ADM), a multifunctional regulatory peptide, on sugar sensing in mice since it affects the expression of SGLT1 in rat enterocytes. We found that ADM receptor components were expressed in T1R3-positive taste cells. Analyses of chorda tympani (CT) nerve responses revealed that ADM enhanced responses to sugars but not to artificial sweeteners and other tastants. Moreover, ADM increased the apical uptake of a fluorescent D-glucose derivative into taste cells and SGLT1 mRNA expression in taste buds. These results suggest that the T1R-independent sweet taste pathway in mouse taste cells is a peripheral target of ADM, and the specific enhancement of gustatory nerve responses to sugars by ADM may contribute to caloric sensing and food intake.

## 1. Introduction

The gustatory system plays an important role in evaluating chemical contents in foods. In particular, sweet taste, one of the preferable sensations, is dedicated to detecting carbohydrates for caloric intake. Not only sugars (such as glucose and sucrose) but also artificial sweeteners (such as SC45647, saccharin, aspartame, and cyclamate), sweet amino acids, and even some proteins elicit sweet tastes. All these sweeteners activate taste receptor family 1 members 2 and 3 (T1R2/T1R3), members of family C of the G-protein-coupled receptors (GPCRs). Thus, T1R2/T1R3 function as sweet taste receptors [1,2], and T1R-dependent mechanisms may be dominant for the detection of various sweeteners. However, mice lacking the T1R-dependent pathway (T1R3-knockout (KO), gustducin-KO, or transient receptor potential melastatin 5 (TRPM5)-KO mice) still showed gustatory responses to multiple sugars, particularly glucose, suggesting the involvement of other detection systems for sugars on the tongue [3]. Gurmarin, a highly specific sweet-taste-suppressing protein in rodents, was reported to bind to T1R3, resulting in the inhibition of T1R2/T1R3 activation [4]. In single gustatory nerve fiber recordings from mice, sweet-sensitive fibers were divided into two groups: gurmarin-sensitive and gurmarin-insensitive types [3]. These data also imply the existence of T1R-independent receptor systems for sugar detection in mice.

Considering the phenotypical similarity between intestinal cells and taste receptor cells, glucose transporters have been considered T1R-independent sugar sensors in taste cells. Glucose transporters (GLUTs), sodium–glucose cotransporter 1 (SGLT1), and two components of the ATP-gated K^+^ (K_ATP_) metabolic sensor (sulfonylurea receptor 1 (SUR1) and potassium inwardly rectifying channel 6.1) were found to be expressed in taste cells [5,6,7], suggesting that these molecules could contribute to sugar sensing in taste cells. In addition, the functional significance of SGLT1 in sugar sensing on the tongue was demonstrated via gustatory nerve recordings, single-fiber recordings, and the apical uptake of the fluorescent D-glucose derivative 2-(N-(7-nitrobenz-2-oxa-1,3-diazol-4-yl)amino)-2-deoxyglucose (2-NBDG) into taste cells [8]. After the uptake of glucose into taste cells, ATP is produced and inhibits the K_ATP_ channel to induce cell depolarization; thus, taste cells are excited by glucose via a glucose-transporter-dependent pathway. In addition, other sugars are able to activate this pathway because sweet taste cells express the disaccharide-digesting enzyme α-glucosidase, and the pharmacological inhibition of this enzyme reduced mouse taste nerve responses to disaccharide [9]. Collectively, T1R-independent and T1R-dependent pathways may function independently of each other. Interestingly, the oral administration of glucose and sucrose but not artificial sweeteners induced cephalic-phase insulin release (CPIR), a pulse of insulin elicited within minutes of sensory stimulation, in in wild-type and T1R3-KO mice [10,11]. Such CPIR was not observed in SUR1-KO mice [11], suggesting that a T1R-independent, K_ATP_-channel-dependent sugar detection mechanism plays an important role in the induction of CPIR. These data also imply different roles of T1R-dependent and T1R-independent sugar detection systems on the tongue in the regulation of animal behaviors and homeostasis.

There is growing evidence that taste function can be modulated by hormones or other factors that act on the receptors present in taste receptor cells. In the case of sweet taste, the anorexigenic mediator leptin selectively suppresses sweet taste responses, and this effect may be mediated by its receptor Ob-Rb, which is expressed in T1R3-positive taste cells [12,13,14,15]. Conversely, endocannabinoids, which are orexigenic mediators, selectively enhanced sweet taste responses via cannabinoid receptor type 1 (CB1), which is expressed on T1R3-positive taste cells [16,17]. Interestingly, leptin and endocannabinoids are reported to function on T1R-dependent and T1R-independent sweet taste pathways independently and may contribute to energy intake by modulating sweet taste sensitivities competitively [17].

Here, we focused on adrenomedullin (ADM) as another sweet-modulating hormone. ADM, a peptide hormone first discovered in 1993 in a panel of peptides extracted from a human pheochromocytoma [18], is widely expressed in different tissues [19] and is now known to play multiple roles in vasodilatory, hypotensive effects and in regulating hormone secretion, inflammatory responses, and glucose metabolism [20]. ADM and the calcitonin gene-related peptide (CGRP) share the same receptor and primarily activate the calcitonin receptor-like receptor (CRLR). ADM-specific receptors exist as a complex of CRLR and receptor-activity-modifying protein 2 (RAMP) or RAMP3 [21]. In the oral cavity, ADM was reported to be present in saliva secreted from oral keratinocytes and cells of the salivary glands [22], and RNA sequencing techniques revealed the expressions of CRLR and RAMPs in the taste cells of mouse taste buds [23,24]. However, the potential role of ADM in the oral cavity remains unknown. Intriguingly, ADM was reported to increase the expression of SGLT1 protein and the uptake of the nonmetabolizable glucose derivative α-methylglucoside in the brush-border membranes of rat enterocytes [25]. Moreover, mRNA expressions of CRLR and RAMPs were found in TRPM5-positive taste cells [23] and T1R3-positive taste cells [24] of mouse taste buds. Therefore, we hypothesized that like leptin and endocannabinoids, ADM may have some modulatory role in sweet taste responses, especially those mediated by the T1R-independent pathway.

In this study, we first examined the expression of ADM receptors in mouse taste buds through the use of immunohistochemistry. We then investigated the potential effects of ADM on the glucose transporter system in taste cells by analyzing mouse chord tympani (CT) nerve responses to various tastants, the apical uptake of 2-NBDG into taste cells, and mRNA expression levels of the sweet taste receptor component T1R3 and glucose transporters before and after the administration of ADM. To our knowledge, no reports thus far have focused on the differences in function between the T1R-dependent and T1R-independent pathways. This is the first study to investigate the regulating factor that selectively modulates newly found sweet taste pathways via SGLT1 in mouse taste buds. The elucidation of functional differences between these sweet taste pathways will shed new light on the development of anti-obesity treatments.

## 2. Materials and Methods

All experimental procedures in this study were performed in accordance with the National Institutes of Health Guide for the Care and Use of Laboratory Animals and were approved by the Committees for Laboratory Animal Care and Use at Kyushu University, Japan (approval numbers A19-179-1 (Research and Development Center for Five-Sense Devices) and A21-360-2 (Section of Oral Neuroscience, Graduate School of Dental Science)).

### 2.1. Animals

The subjects were adult male C57BL/6J (B6) (Charles River; *n* = 58) mice, T1R3 green fluorescent protein (GFP) [26] (*n* = 3) mice, and T1R3-KO (*n* = 14) [27] mice. The T1R3-GFP mice and T1R3-KO mice were originally derived from the C57BL/6J strain at Mount Sinai Medical School via homologous recombination in C57BL/6 embryonic stem cells and were maintained in this genetic background (age: 8–16 weeks; weight: 20–32 g). We used male mice to avoid the effects of sex differences because the ADM gene is reported to be directly activated by estrogen receptors in mice [28].

### 2.2. Immunohistochemistry

Immunohistochemistry was conducted as described previously [16]. Tongues from B6 and T1R3-GFP mice (8–12 weeks old and weighing 25.7 ± 1.9 g) were dissected and fixed in a 4% paraformaldehyde (PFA) solution in phosphate-buffered saline (PBS) for 50 min. Following dehydration in sucrose solution (10% for 1 h, 20% for 1 h, and 30% for 3 h at 4 °C), the fixed tissue block was embedded in an optimal cutting temperature (OCT) compound (Sakura Finetek, Tokyo, Japan), and 10 µm thick sections were obtained and mounted on silane-coated glass slides. The tongue slices were subject to antigen retrieval via incubation in Histo VT One (Nakalai Tesque, Kyoto, Japan) for 20 min at 80 °C, followed by a 1 h incubation in Blocking One solution (Nacalai Tesque). Subsequently, the sections were incubated overnight at 4 °C with the primary antibodies. After several rinses with a Tris-NaCl-Tween (TNT) buffer, the slides were incubated in secondary antibodies for 2 h and then washed again. Immunofluorescence and GFP fluorescence were visualized with the use of a laser scanning microscope (FV-1000; Olympus), and the images were captured using Fluoview (Olympus, Tokyo, Japan). The primary antibodies used in this study were anti-GFP (1:1000; chicken anti-GFP, cat. no. GFP-1020, Aves Labs, Inc., Davis, CA, USA), anti-carbonic anhydrase 4 (CA4) (1:100; goat anti-CA4, cat. no. AF2414, R&D Systems, Minneapolis, MN, USA), anti-CRLR (1:300; rabbit anti-T1R3, cat. no. bs-1860R, Bioss Antibodies, Woburn, MA, USA), and anti-RAMP2 (25 µg/mL; rat anti-RAMP2, cat. no. MAB6500, R&D Systems, Minneapolis, MN, USA). The secondary antibodies used were as follows: for GFP (1:300; goat anti-chicken IgY (H+L) secondary antibody, Alexa Fluor 488, cat. no. A10039, Invitrogen, Waltham, MA, USA), for CA4 (1:300; Alexa Fluor 488 donkey anti-goat IgG, cat. no. A11055, Invitrogen), for CRLR (1:300; Alexa Fluor 568 donkey anti-rabbit IgG, cat. no. A40081, Invitrogen), and for RAMP2 (1:300; Alexa Fluor 647 donkey anti-rat IgG, cat. no. A31571, Invitrogen). The quantification of positive cells was performed in the taste buds located in the fungiform papillae (FP).

### 2.3. Chorda Tympani Nerve Recording

We recorded the whole nerve responses to taste substances applied to the mouse tongue from the chorda tympani (CT) nerve, using the methodology described in previous studies [12,29]. Each mouse was anesthetized with 50–60 mg/kg b.w. pentobarbital and secured in the supine position. After tracheal cannulation, the CT nerve on the right side was severed from the surrounding tissues after resecting the pterygoid muscle and cutting where it entered the bulla. The entirety of the isolated nerve was then placed on the Ag/AgCl electrode, while an indifferent electrode was attached to nearby tissue. We utilized an ER-1 Differential Extracellular Amplifier (ER-1 Differential Extracellular Amplifier, Cygnus Technology, Delaware Water Gap, PA, USA) to amplify the neural activities’ inputs. Signal monitoring was carried out using an oscilloscope and an audio monitor. The responses of the whole nerves were integrated and recorded on a computer by the integrator (Yutaka Electronics Company, Gifu, Japan) with a time constant of 1.0 s and the Power Lab system (Power Lab/sp4, AD Instruments, Bella Vista, NSW, Australia). To appropriately stimulate the FP with taste solutions, a silicone rubber flow chamber was used to pour the taste solution onto the anterior half of the tongue. Taste solutions (100 mM of NH_4_Cl, 10–1000 mM of NaCl, 100 mM of KCl, 10–1000 mM of glucose (Glc) with and without 10 mM of NaCl, 500 mM of sucrose (Suc) with and without 10 mM of NaCl, 1 mM of SC45647 (SC) with and without 10 mM of NaCl, 10 mM of HCl, 20 mM of quinine-HCl (QHCl), and 100 mM of monopotassium glutamate (MPG)) were used in this study. The taste solutions were supplied to the tongue utilizing the flow of gravity and flushing the mouse tongues for 30 s. Distilled water was used to wash out the tongues for an interval of ~1 min between successive stimulations. After confirming recovery to at least 85% of the control level of response for most cases of the whole nerve recordings, these series of simulations were repeated.

After a series of control responses were recorded, 0–10^−4^ mol/kg b.w. of ADM (Peptide Institute, Osaka, Japan) was administered intravenously through the femoral vein using a microsyringe pump (ESP-32, Eicom, Kyoto, Japan) at a constant speed of 60 µL/min for a duration of 1 min. To examine the effect of ADM receptor inhibition, each mouse was administrated a single intravenous (i.v.) injection of the adrenomedullin receptor antagonist AM (22–52) (22–52-adrenomedullin; 0–10^−5^ mol/kg b.w.; Anapsec, Fremont, CA, USA) 5 min before the onset of recording with ADM. ADM and AM (22–52) were dissolved in physiological saline.

The recording of taste responses was continued until a particular time point (0, 5, 10, 30, 60, 90, and 120 min after injection). The criteria for a stable recording was the magnitude of the response to 100 mM of NH_4_Cl at the beginning and end of each stimulus series, defined as a deviation within 15%, and we used only responses from stable recordings for the data analysis. In the analysis of whole nerve responses, we used the magnitudes of the integrated whole nerve responses recorded 5, 10, 15, 20, and 25 s after the beginning of stimulation and then averaged and normalized these values to the magnitudes of the responses to 100 mM of NH_4_Cl to account for individual variations in absolute responses. After the completion of the series of experiments, each mouse was euthanized via an anesthesia overdose. All experiments were conducted at room temperature (24–25 °C).

### 2.4. Uptake of a Fluorescent D-Glucose Derivative into Taste Bud Cells

We recorded the uptake of a fluorescent D-glucose derivative into the taste cells of mouse FP via the previously described methods [8]. The subjects were adult (>8 weeks old) male B6 mice (*n* = 15; *n* = 3 for each condition). At 25 min prior to euthanasia by cervical dislocation, AM (22–52) or saline was injected intravenously, and ADM or saline was administered intravenously 5 min later. After cervical dislocation, the anterior part of the tongue was removed and washed with Tyrode’s solution (NaCl, 140 mM; KCl, 5 mM; CaCl_2,_ 1 mM; MgCl_2_, 1 mM; NaHCO_3_, 5 mM; glucose, 10 mM; sodium pyruvate, 10 mM; HEPES, 10 mM; pH 7.4 adjusted with NaOH). The apical side of the removed tongue was wiped with a KimWipe (Nippon Paper Crecia, Tokyo, Japan) to remove adhering solution and then treated with 10 mM 2-NBDG + 10 mM NaCl with or without 1 mM of phlorizin for 15 min at room temperature in a dark chamber. When the phlorizin pretreatment was used, the tongue was exposed to 1 mM of phlorizin for 10 min prior to the treatment with 2-NBDG. Subsequently, the treated tongue was rinsed with Tyrode’s solution and gently wiped with a cotton swab. A section of tongue muscle tissue was then extracted using forceps and carefully spread out in a culture dish coated with Sylgard, with the apical side facing upward. The pinned tongue tissue was incubated in 4% paraformaldehyde in PBS for 5 min at the temperature of 4 °C and subsequently rinsed several times with Tyrode’s solution. To obtain optical sections of the taste buds located in the fungiform papillae (FP), a laser scanning confocal microscope (FV-1000 and Fluoview) equipped with water-immersion objective lenses (LUMPlanFl × 40, Olympus) was employed. The imaging process used the following settings: excitation 488 nm, emission 500–600 nm, section thickness of 1.5 µm, and a total of 15 sections. Fluorescent images of the 10th optical section, located 15 µm below the apical side, were subjected to analysis using ImageJ software (ver.1.51, NIH, Bethesda, MD, USA). Approximately 20–30 taste buds were analyzed per animal.

### 2.5. RT-qPCR

B6 mice were subjected to isoflurane anesthesia and euthanized via cervical dislocation. AM (22–52) or saline was injected intravenously 25 min before cervical dislocation, and then ADM or saline was administered intravenously 5 min later. Elastase treatment (0.5–1 mg/mL; Elastin Products, Owensville, MO, USA; incubated for 10 min at room temperature) was performed on the epithelial tissues of the anterior and posterior parts of the tongue, allowing for the subsequent removal of the peeled tissue and the surrounding tissue. Using a glass pipette, the taste buds in the fungiform papillae (FP) (50 taste buds per mouse) were collected in Tyrode’s solution. Under a microscope, the circumvallate papilla (CV) tissue was isolated from the surrounding tissue, and the Von Ebner’s glands were removed. RNA extraction from mouse taste tissues was carried out using a Fast Gene^™^ RNA Premium Kit (cat. no. FG-81050, Nippon Genetics, Japan), and the RNA samples were quantified using an LSM Nano Drop ND-1000 (Thermo Fisher Scientific, Waltham, MA, USA). The cDNA synthesis was performed using Super Script VILO Master Mix (cat. no. 11755050, Thermo Fisher Scientific). An RT-qPCR was performed as follows: 15 min at 95 °C (1 cycle); 15 s at 95 °C, 30 s at 60 °C, and 30 s at 72 °C (40 cycles). Each 10 µL of PCR solution contained 5 µL of Fast SYBR Green Master Mix (cat. no. 4385612, Fisher Scientific), 0.2 µL of cDNA solution, 0.5 µL of primer, and 4.3 µL of double-distilled water (StepOnePlus, Applied Biosystems, Foster City, CA, USA). StepOne Software (ver. 2.3, Applied Biosystems) was utilized for data analysis. We performed each assay in duplicate, and the runs were replicated three times. All results of the RT-qPCR were normalized using the ΔΔCt method with glyceraldehyde-3-phosphate dehydrogenase (Gapdh) in each sample as a reference [30]. For each gene, primer pairs were selected in such a way that each primer corresponded to a different exon. The primers used for each gene are shown in Table 1.

### 2.6. Data Analysis

The effects of ADM and AM (22–52) in B6 mice and T1R3-KO mice were evaluated via Student’s *t*-test. The effects of ADM and AM (22–52) on CT nerve responses and the time-dependent effects of ADM and AM (22–52) were analyzed via a two-way ANOVA followed by a post hoc Student’s *t*-test with Bonferroni correction. The dose-dependent effects of ADM and AM (22–52), the effects of ADM and AM (22–52) on synergistic responses to binary mixtures of tastants and 10 mM of NaCl and on 2-NBDG uptake in mouse taste cells, and the mRNA expression levels of sweet taste receptor components and glucose transporters were evaluated using a one-way ANOVA followed by a post hoc Tukey’s HSD test or Student’s *t*-test. The calculations were performed using the statistical software package IBM SPSS Statistics (ver.19, IBM, Armonk, NY, USA).

## 3. Results

### 3.1. Immunohistochemistry

First, we analyzed the expressions of CRLR and RAMP2 in the FP taste buds of B6 mice. We used two cell type markers, the T1R3 sweet and umami receptor component for type II cells [31] and CA4 for type III cells [32], to investigate the co-expression patterns of CRLR, RAMP2, and these cell-type markers in FP.

Immunoreactivities for CRLR and RAMP2 were detected in some FP taste cells. CRLR immunoreactivity was observed in both T1R3-positive cells (T1R3/CRLR: 56.3%) and CA4-positive cells (CA4/CRLR: 36.1%). On the other hand, most RAMP2-immunopositive cells expressed T1R3 (T1R3/RAMP2: 87.5%) but not CA4 (CA4/RAMP2: 11.8%). In summary, the majority of taste cells expressing both CRLR and RAMP2 (ADM receptor) were immunopositive for T1R3 (T1R3/ADM receptor: 89.6%), but few of them expressed CA4 (CA4/ADM receptor: 12.2%) (Figure 1). The inverse co-expression ratios are shown in Table 2. RAMP2 and CRLR signals were detected in the myocardium of the ventricle, consistent with a previous study [33].

### 3.2. CT Nerve Recordings

To elucidate the function of ADM in peripheral taste tissue, we examined the effect of ADM on gustatory nerve responses in B6 mice. CT nerve responses to Glc and Suc after an i.v. injection of 10^−6^ mol/kg b.w. of ADM were significantly greater than those after an i.v. injection of the vehicle (Figure 2A,B, Table 3). The relative responses to Glc and Suc with the ADM injection were about 1.5 and 1.6 times greater than those with the vehicle injection. In contrast, the CT nerve responses to other tastants such as SC, NaCl, KCl, HCl, QHCl, and MPG were not significantly different between ADM and vehicle injections (Figure 2A,B, Table 3). The ADM injection had no effect on the CT nerve response to the artificial sweetener SC, suggesting that the effect of ADM may be specific to sugar responses (Figure 2A,B, Table 3).

ADM significantly enhanced CT nerve responses to 300–1000 mM Glc (Figure 2C, Table 4). This effect was observed approximately 5 min after an i.v. injection of ADM, reached a near-maximum level (about 150% of control) approximately 10 min after the injection, and returned to the control level 60 min after the injection (Figure 2D, Table 5). The effect of ADM on sweet response was dose-dependent and saturated at 10^−6^ mol/kg b.w. (Figure 2E; Glc: *F*_(5,36)_ = 28.2, *p* < 0.001; NaCl: *F*_(5,36)_ = 0.318, *p* = 0.899). In contrast, ADM had no such effect on responses to NaCl (Figure 2C–E, Table 4 and Table 5).

Next, we examined whether the ADM receptor mediates the effect of ADM on sugar responses, using the ADM receptor antagonist AM (22–52). An enhancement of CT nerve responses to sugars was not observed in the B6 mice pretreated with AM (22–52) (Figure 3A,B, Table 3). At a dose of 10^−7^ mol/kg b.w. of AM (22–52), CT nerve responses to 300–1000 mM Glc were similar between the ADM and vehicle injection groups (Figure 3C, Table 6). The enhancement of CT nerve responses to 500 mM of Glc 5–30 min after an ADM injection was blocked by the pretreatment with AM (22–52) (Figure 3D, Table 7). AM (22–52) significantly blocked the enhancement of the CT nerve response to Glc at ≥10^−7^ mol/kg b.w. (Figure 3E; Glc: *F*_(5,36)_ = 21.9, *p* < 0.001). On the other hand, AM (22–52) had no effect on responses to NaCl (Figure 3C–E; NaCl: *F*_(5,36)_ = 1.06, *p* = 0.401 (Figure 3E), Table 6 and Table 7). Pretreatment with AM (22–52) had no effect on responses to the artificial sweetener SC or other tastants (Figure 3B).

ADM enhanced CT nerve responses to sugars but not to artificial sweeteners. Therefore, ADM may enhance responses to sugars, an effect which might be mediated by T1R-independent sweet taste pathways. This was tested by using T1R3-KO mice. In these mice, ADM injection also significantly enhanced CT nerve responses to Glc and Suc but not to SC or other tastants (Figure 4A, Table 3). These enhancing effects of ADM in T1R3-KO mice were blocked by pretreatment with AM (22–52) (Figure 4B, Table 3). Treatment with ADM plus AM (22–52) pretreatment had no effect on responses to SC or other tastants in T1R3-KO mice (Figure 4A,B, Table 3).

The addition of 10 mM of NaCl to sugars but not artificial sweeteners or other tastants elicited synergistic CT nerve responses in both B6 and T1R3-KO mice. This effect was abolished after a pretreatment with phlorizin, a competitive inhibitor of SGLT1, indicating that SGLT1 contributed to the synergistic responses to the mixtures [8]. ADM enhanced synergistic responses to binary mixtures of Glc or Suc with 10 mM of NaCl but not to a mixture of SC with 10 mM of NaCl in T1R3-KO mice (Figure 5A,B). AM (22–52) inhibited such enhancing effects of ADM (Figure 5B; Glc: *F*_(5,44)_ = 24.3, *p* < 0.001; Suc: *F* _(5,36)_ = 10.5, *p* < 0.001; SC: *F* _(5,36)_ = 0.402, *p* = 0.844). The enhancing effects of ADM on responses to Glc and Suc and mixtures of these sugars with 10 mM of NaCl were abolished after phlorizin treatment of the tongue (Figure 5C; Glc: *F* _(3,24)_ = 1.65, *p* = 0.205; Suc: *F* _(3,28)_ = 0.911, *p* = 0.448; SC: *F* _(3,28)_ = 0.068, *p* = 0.976). Taking these data together, we hypothesized that a sugar-sensing pathway via SGLT1 but not via the T1Rs on the tongue might be the target for ADM to enhance sugar responses.

### 3.3. 2-NBDG Uptake in Taste Cells

2-NBDG can enter the cell through glucose transporters such as GLUTs and SGLTs [34]. Following apical treatment of the tongue with 2-NBDG + 10 mM NaCl, some taste cells in the FP taste buds clearly exhibited 2-NBDG fluorescence (Figure 6A). The fluorescence of 2-NBDG in the taste cells was significantly enhanced by the injection of ADM (Figure 6A,B). The AM (22–52) pretreatment inhibited the enhancing effect of ADM on 2-NBDG uptake in taste cells. In addition, pretreatment with phlorizin on the tongue reduced 2-NBDG uptake and abolished the enhancing effect of ADM on 2-NBDG uptake in taste cells (Figure 6B, *F*_(4,302)_ = 43.7, *p* < 0.001). These results indicate that ADM might increase glucose uptake via SGLT1 expressed in apical-side taste cells.

### 3.4. RT-qPCR

ADM was reported to increase the expression of SGLT1 and the uptake of a nonmetabolizable glucose derivative in rat intestines [25]. Therefore, we examined the effect of ADM injection on the expression levels of mRNAs for glucose transporters in mouse FP and CV taste buds (Figure 7, Table 8). ADM enhanced SGLT1 mRNA expression significantly both in FP and in CV taste buds. On the other hand, ADM injection had no effects on the mRNA expressions of other glucose transporters (GLUT2, GLUT4, and GLUT8) or T1R3 in taste cells [6,7]. Consistent with a previous study using rats [25], ADM increased SGLT1 mRNA expression in mouse small intestines. GLUT2, GLUT4, GLUT8, and T1R3 mRNA expressions in the small intestine were not affected by an ADM injection. AM (22–52) pretreatment significantly blocked the enhancement of SGLT1 mRNA expression in taste cells and the small intestine by ADM. AM (22–52) had no effects on the mRNA expressions of other glucose transporters or T1R3 (Figure 7, Table 8).

## 4. Discussion

We found the expression of CRLR in both T1R3-positive cells and in CA4-positive cells. On the other hand, most RAMP2-expressing taste cells were T1R3-positive but not CA4-positive (Figure 1, Table 2). This is consistent with previous RNA sequencing data showing the expression of CRLR in both type II and in type III cells and the expression of RAMP2 in the type II cells of mouse taste buds [23,24]. These results indicate that at the very least, an ADM receptor consisting of CRLR and RAMP2 is preferentially expressed in T1R3-positive taste cells. CRLR and RAMP2 or RAMP3 form the ADM receptor, whereas CRLR and RAMP1 form the CGRP receptor [35]. The expression of CGRP was reported in the sensory neuron innervating the CV in rats [36]. Furthermore, CGRP evoked serotonin (5-HT) secretion from presynaptic cells (type III taste cells) and released 5-HT-provided negative paracrine feedback onto receptor cells (type II taste cells) via the activation of 5-HT_1A_ receptors, subsequently inducing a reduction in ATP secretion [37]. Therefore, the CRLR expressed on CA4-positive taste cells may function as a component of the CGRP receptor. This should be investigated in future studies.

The intravenous injection of ADM enhanced CT nerve responses to sugars but not to artificial sweeteners in B6 mice (Figure 2A–D). The half-life of circulating ADM is reported to be 16~22 min [38]. Consistent with this report, the enhancing effect of ADM on CT nerve responses to sugars was observed from 5 min after an i.v. injection of ADM, reached its maximum level approximately 10 min after, and was maintained until 30 min after, when the responses started to recover (Figure 2B). A similar time course of the effect of ADM after an i.v. injection was reported in *in vivo* experiments investigating its effects on vasodilation [39] and cardiovascular changes [40] in rats. The enhancing effects of ADM on CT nerve responses to sugars were reduced via pretreatment with the ADM receptor antagonist AM (22–52) (Figure 3A–D), indicating that these enhancing effects of ADM were mediated by ADM receptors. In our experiments, CRLR and RAMP2 proteins were likely to be expressed in T1R3-positive taste cells, which may play a key role in detecting sugars in the oral cavity. Therefore, ADM may affect T1R3-positive cells expressing both CRLR and RAMP2 to enhance the sugar responses of these cells.

Enhancing effects of ADM on responses to sugars were also found in T1R3-KO mice (Figure 4A,B). Moreover, the SGLT1 inhibitor phlorizin reduced the enhancing effects of ADM on sugars and sugar-NaCl mixtures in T1R3-KO mice (Figure 5A,B). We had reported that SGLT1 may mediate the T1R-independent sweet taste of sugars [8]. Taking these data together, SGLT1 may be a key component for the enhancement of sugar responses by ADM. To support this hypothesis, the apical uptake of a fluorescent D-glucose derivative in taste cells was also increased by ADM treatment, and these effects were almost completely blocked by AM (22–52) or phlorizin pretreatment (Figure 6). ADM injection also increased the mRNA expression level of SGLT1 but not of other glucose transporters or T1R3 in FP and CV taste buds, and AM (22–52) inhibited this effect (Figure 7).

How does ADM modulate SGLT1 expression and increase the uptake of glucose into taste cells? In many types of cells, both ADM and CGRP receptors are coupled with the stimulatory G protein G_αs_, which activates adenylate cyclase and increases intracellular cyclic adenosine monophosphate (cAMP) levels [41]. Taste cells are also reported to express several isoforms of adenylyl cyclase [42] and phosphodiesterase [43], which synthesize and degrade cAMP, respectively. Cyclic AMP has been shown to increase the amiloride-sensitive Na^+^ current in isolated hamster taste bud cells [42] and the CT nerve response to NaCl in rats [44]. These enhancements may occur via cAMP pathway activation, leading to the recruitment of an intracellular pool of the epithelial sodium channel (ENaC) to the plasma membrane [44]. Angiotensin II (Ang II) is a well-known vasopressor hormone, and its receptor is also a G-protein-coupled receptor (GPCR). Ang II stimulates early proximal tubule bicarbonate absorption by decreasing the intracellular cAMP level via Ang II receptor activation in rats [45]. On the other hand, in the kidney, aldosterone (Aldo), also a vasopressor hormone, binds to the cytosolic mineralocorticoid receptor, and the Aldo/receptor complex is then translocated to the nucleus, leading to ENaC mRNA transcription and then ENaC protein trafficking to the cell surface [46]. Furthermore, cAMP and serum and glucocorticoid-inducible kinase, a downstream molecule of Aldo, is reported to increase ENaC protein trafficking [47]. We previously reported that Ang II reduced taste sensitivities for NaCl in mice [48]. Conversely, it was demonstrated that Aldo enhanced taste sensitivities for NaCl in rats [49]. These reductions and enhancements of responses to NaCl may have occurred via the modulation of cAMP through the activation of each hormone’s receptors [48,49], resulting in the downregulation or upregulation of ENaC on the plasma membrane. Focusing on SGLT1, the activation of the cAMP-PKA pathway upregulated SGLT1 expression in the intestines of sheep [50,51]. In rat intestines, ADM increased SGLT1 protein expression and α-methylglucoside absorption [25]. Therefore, the activation of the cAMP pathway via ADM receptor may upregulate SGLT1 expression. Taken together, such mechanisms may be involved in the enhancement of CT nerve responses to sugars (Figure 2D), 2-NBDG uptake (Figure 6), and SGLT1 mRNA expression levels (Figure 7) within 30 min of ADM administration. Although there were many reports demonstrating the contributions of the cAMP-PKA pathway to the regulation of SGLT1 expression, its time course varied greatly from 20 min [52] to several hours [53]. These differences may result from differences in the cell lines used, in vivo vs. in vitro approaches, or experimental designs. To clarify the possibility of SGLT1 recruitment via ADM receptor activation in taste cells, further studies are needed.

At the peripheral taste tissue, multiple sugar sensors and information pathways may play differential roles that work in combination to support food intake and energy homeostasis. For example, metabolic sensor K_ATP_ channels are coupled with the glucose transporter pathway in taste cells and play a crucial role in CPIR after oral stimulation with sugars [10,11]. The anorexigenic mediator leptin suppressed sweet taste sensitivities, and its suppression of sweet responses was mediated by the leptin receptor Ob-Rb [12,13,14,15] and by phosphoinositide 3-kinase in the K_ATP_ channel pathway, which is expressed on T1R3-positive taste cells [54]. Conversely, 2-arachidonylglycerol (2-AG), one of the endocannabinoids known as orexigenic mediators, selectively enhanced sweet taste responses via the expression of CB1 on T1R3-positive taste cells [16]. Interestingly, 2-AG synthase diacylglycerol lipase α and β were also expressed in T1R3-positive taste cells in mice [17]. The major pathway of 2-AG synthesis is reported to be the GPCR–phospholipase C–diacylglycerol pathway [55,56]. These data imply that a T1R-dependent pathway may receive autocrine feedback via 2-AG [17]. Circulating leptin but not endocannabinoids may act as a dominant modulator for sweet taste in lean wild-type mice [17]. Therefore, endocannabinoids may function as more effective sweet taste modulators under the conditions of deficient leptin signaling by increased 2-AG synthesis in taste tissue [17]. Leptin and endocannabinoids may contribute to energy intake by modulating sweet taste sensitivities competitively via T1R-dependent and T1R-independent sweet taste signaling pathways [17]. Focusing on the neural taste pathway, single-fiber recordings of CT nerve responses in wild-type mice revealed that S-type fibers showing maximal responses to sucrose could be classified into T1R-dependent (phlorizin-insensitive), Glc, and mixed (SGLTs and T1Rs) types [8]. Interestingly, a subset of S-type fibers also respond to glutamate with inosine-5′-monophosphate [57] and to the long-chain fatty acids linoleic acid and oleic acid [58]. Therefore, there is a group of fibers that respond to three major nutrients, suggesting that the taste cells connecting to this subset of S-type fibers may function to find energy sources [58]. Taking into consideration the modulation systems of sweet taste sensitivity between T1R-dependent and/or T1R-independent pathways, as mentioned above, ADM may also contribute to caloric sensing and food intake via the T1R-independent sweet taste pathway. Interestingly, an intracerebroventricular injection of ADM has been proposed to regulate food intake in previous studies with animal models [59,60,61]. However, this should be verified by future studies because detailed mechanisms within and between peripheral and central taste sensing systems remain unresolved.

In conclusion, we have demonstrated that an SGLT1-dependent sugar taste pathway may be the target of ADM for the enhancement of sugar information. ADM receptor components were expressed on T1R3-positive taste cells in mouse FP. An ADM injection enhanced CT nerve responses to sugars but not to artificial sweeteners or other tastants in both wild-type and in T1R3-KO mice, facilitated glucose uptake in taste cells, and increased SGLT1 mRNA expression. These effects of ADM injection were reduced via a pretreatment with the ADM receptor inhibitor AM (22–52), indicating that these effects of ADM were mediated via the ADM receptor. Moreover, ADM enhanced synergistic responses to binary mixtures of sugars with 10 mM of NaCl, and this effect was blocked by pretreatment with the SGLT1 inhibitor phlorizin, indicating that SGLT1 is a key component for the enhancement of sugar responses by ADM. Taken together, our results indicate that ADM may contribute to caloric sensing and food intake by enhancing gustatory nerve responses to sugars via a T1R-independent pathway that utilizes an SGLT glucose transporter.

## Figures and Tables

**Figure 1 nutrients-15-02941-f001:**
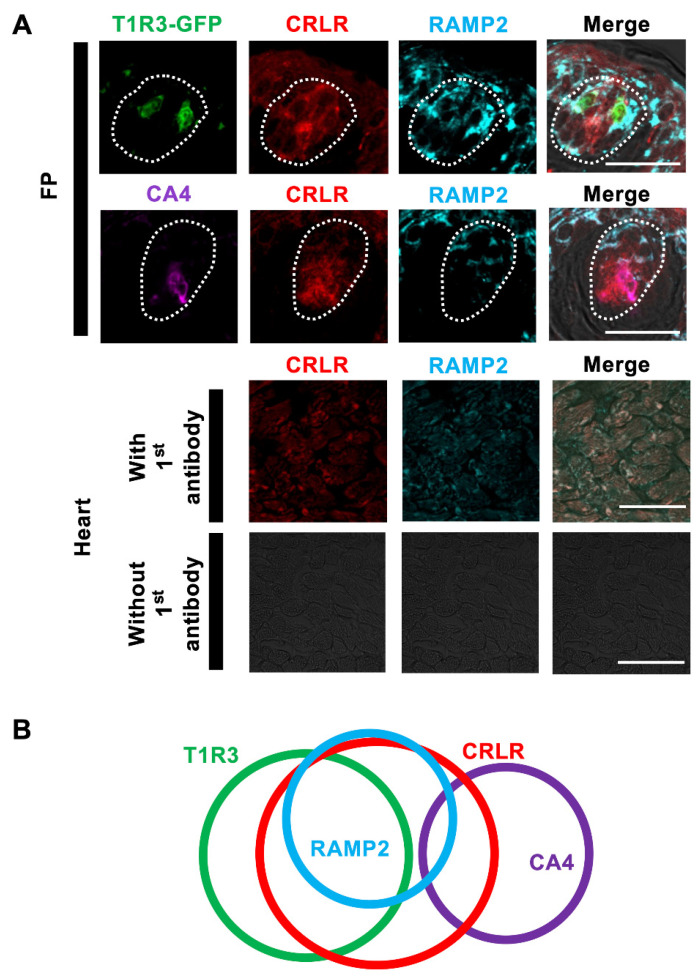
CRLR and RAMP2 immunoreactivities in mouse fungiform papilla (FP) taste cells. (**A**) Co-expression of CRLR (red) and RAMP2 (blue) with T1R3-GFP (green, first row) or CA4 (purple, second row) in FP. Third and fourth rows show CRLR and RAMP2 immunoreactivities in myocardium of heart with or without first antibody. Dotted lines outline the taste bud. Scale bars, 50 µm. (**B**) Summary of the co-expression patterns among CRLR, RAMP2, T1R3, and CA4 in FP taste cells.

**Figure 2 nutrients-15-02941-f002:**
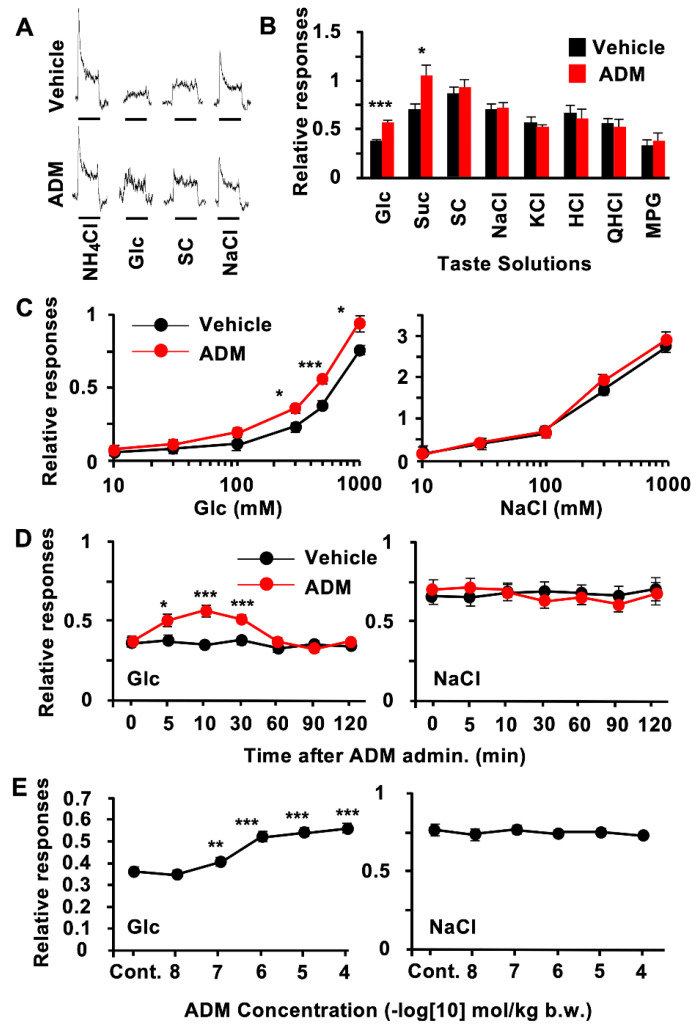
Adrenomedullin (ADM) enhanced chorda tympani (CT) nerve responses to sugars but not to artificial sweeteners and other tastants. (**A**) Typical examples of CT nerve responses of B6 mice 10–30 min after i.v. injection of vehicle (upper traces) or 10^−6^ mol/kg b.w. ADM (lower traces). (**B**) CT nerve responses of B6 mice to 500 mM of glucose (Glc), 500 mM of sucrose (Suc), 1 mM of SC45647 (SC), 100 mM of NaCl, 100 mM of KCl, 10 mM of HCl, 20 mM of quinine HCl (QHCl), and 100 mM of monopotassium glutamate (MPG) 10–30 min after administration of vehicle (black bars) or 10^−6^ mol/kg b.w. of ADM (red bars) (*n* = 7). (**C**), Concentration-dependent CT nerve responses to Glc (left) and NaCl (right) 10–30 min after administration of vehicle (black symbols) or 10^−6^ mol/kg b.w. of ADM (red symbols) (*n* = 7). (**D**), Time-dependent changes in CT nerve responses to 500 mM of Glc (left) and 100 mM of NaCl (right) after administration of vehicle (black symbols) or 10^−6^ mol/kg b.w. of ADM (red symbols) (*n* = 7). (**E**), Dose-dependent effect of ADM treatment on CT nerve responses to 500 mM of Glc (left) and 100 mM of NaCl (right) (*n* = 7). Asterisks show significant differences between vehicle controls (Cont.) and ADM administration (* *p* < 0.05, ** *p* < 0.01, *** *p* < 0.001; Student’s *t*-test (**B**) post hoc *t*-test following repeated two-way ANOVA ((**C**,**D**), with Bonferroni correction) and one-way ANOVA (**E**)). All data are presented as means ± SEMs.

**Figure 3 nutrients-15-02941-f003:**
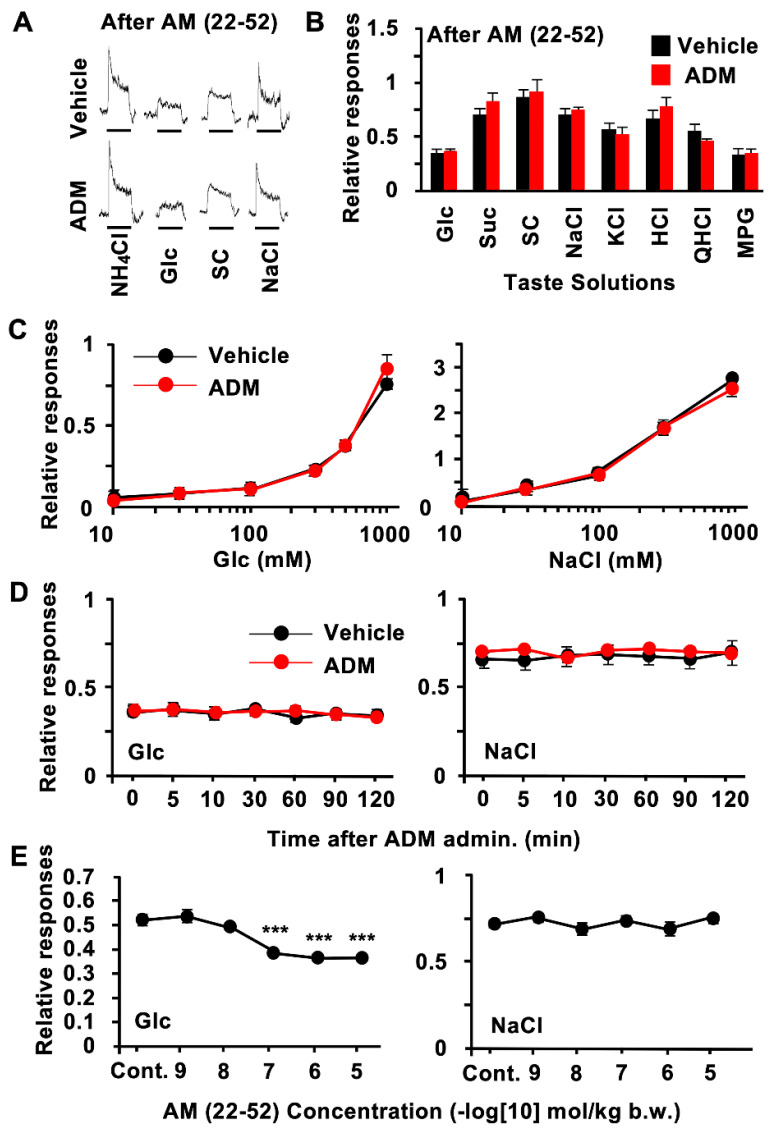
Adrenomedullin (ADM) receptor antagonist AM (22–52) inhibited ADM effect on chorda tympani (CT) nerve responses to sugars. (**A**) Typical examples of CT nerve responses 10–30 min after i.v. injection of vehicle (upper traces) or 10^−6^ mol/kg b.w. of ADM (lower traces) in B6 mice pretreated with 10^−7^ mol/kg b.w. of AM (22–52) intravenously. (**B**), CT nerve responses to 500 mM of glucose (Glc), 500 mM of sucrose (Suc), 1 mM of SC45647 (SC), 100 mM of NaCl, 100 mM of KCl, 10 mM of HCl, 20 mM of quinine HCl (QHCl), and 100 mM of monopotassium glutamate (MPG) 10–30 min after administration of vehicle (black bars) or 10^−6^ mol/kg b.w. of ADM (red bars) in B6 mice pretreated with 10^−7^ mol/kg b.w. of AM (22–52) (*n* = 7). (**C**) Concentration dependence of CT nerve responses to Glc and NaCl 10–30 min after administration of vehicle (black symbols) or 10^−6^ mol/kg b.w. of ADM (red symbols) after pretreatment with 10^−7^ mol/kg b.w. of AM (22–52) (*n* = 7). (**D**) Time-dependent changes in CT nerve responses to Glc and NaCl after administration of vehicle (black symbols) or 10^−6^ mol/kg b.w. of ADM (red symbols) plus pretreatment with 10^−7^ mol/kg b.w. of AM (22–52) (*n* = 7). (**E**), Dose-dependent effect of AM (22–52) pretreatment on enhancing effect of 10^−6^ mol/kg b.w. of ADM on CT nerve responses to 500 mM of Glc (left) and 100 mM of NaCl (right) (*n* = 7). Control (Cont.) shows effects of 10^−6^ mol/kg b.w. of ADM on CT nerve responses to Glc and NaCl after vehicle pretreatment. Asterisks show significant differences between 10^−6^ mol/kg b.w. ADM plus pretreatment with vehicle vs. pretreatment with AM (22–52) (*** *p* < 0.001; post hoc *t*-test following one-way ANOVA (**E**)). Significant differences were not found between vehicle controls and ADM administration after pretreatment with AM (22–52) (**B**–**D**). All data are presented as means ± SEMs.

**Figure 4 nutrients-15-02941-f004:**
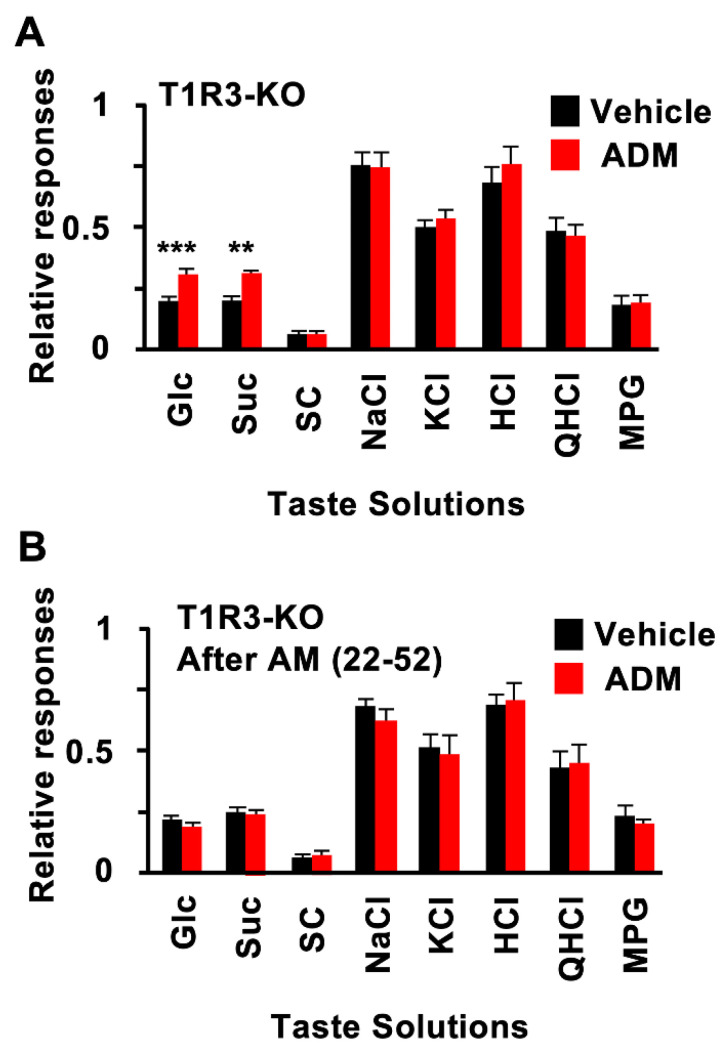
Adrenomedullin (ADM) enhanced chorda tympani (CT) nerve responses to sugars in T1R3-KO mice. (**A**) CT nerve responses of T1R3-KO mice to 500 mM Glucose (Glc), 500 mM sucrose (Suc), 1 mM SC45647 (SC), 100 mM of NaCl, 100 mM of KCl, 10 mM of HCl, 20 mM of quinine HCl (QHCl), and 100 mM of monopotassium glutamate (MPG) 10–30 min after administration of vehicle (black bars) or 10^−6^ mol/kg b.w. of ADM (red bars) (*n* = 7~13). (**B**) CT nerve responses of T1R3-KO mice to Glc, Suc, SC, NaCl, KCl, HCl, QHCl, and MPG 10–30 min after administration of vehicle (black bars) or 10^−6^ mol/kg b.w. ADM (red bars) after pretreatment with 10^−7^ mol/kg b.w. AM (22–52) (*n* = 7~13). Asterisks show significant differences between vehicle controls and ADM administration (** *p* < 0.01, *** *p* < 0.001; Student’s *t*-test (**A**)). After pretreatment with AM (22–52), significant differences were not shown between vehicle controls and ADM administration (**B**). All data are presented as means ± SEMs.

**Figure 5 nutrients-15-02941-f005:**
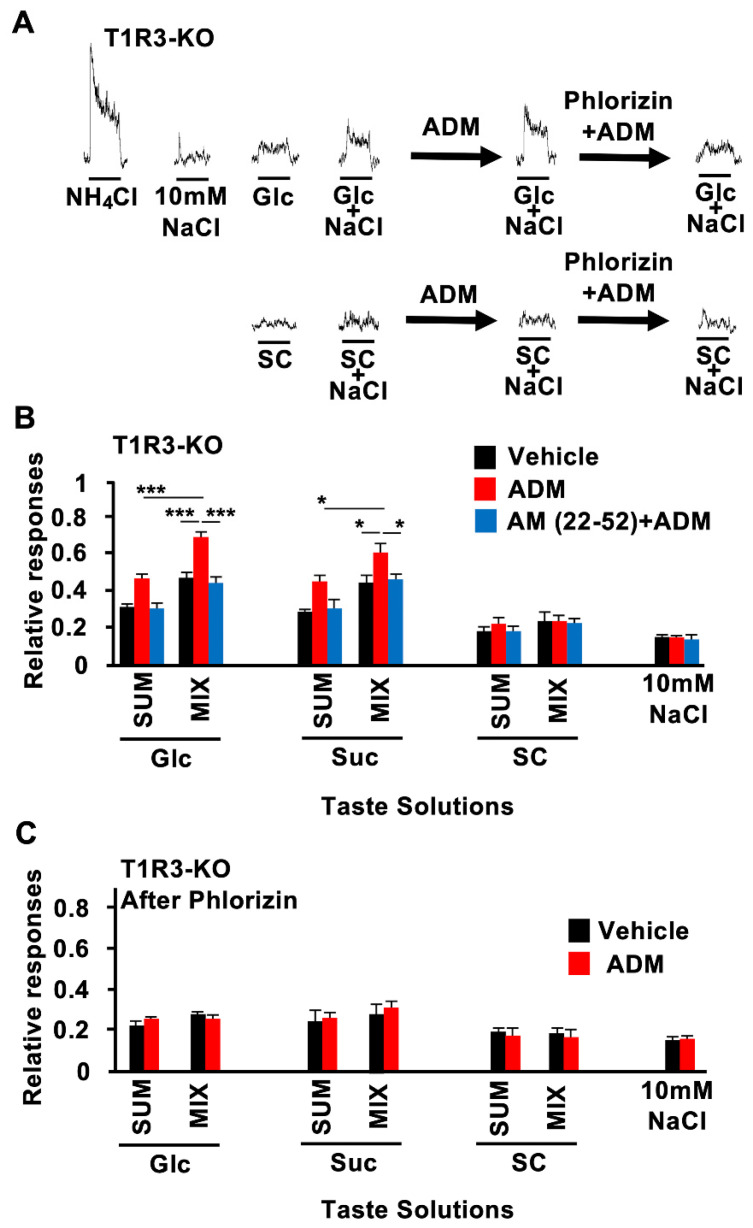
Adrenomedullin (ADM) enhanced synergistic responses to binary mixtures of sugars with 10 mM of NaCl in T1R3-KO mice. (**A**) Typical examples of CT nerve responses 10–30 min after i.v. injection of control (vehicle), 10^−6^ mol/kg b.w. of ADM with vehicle pretreatment (ADM), or 10^−6^ mol/kg b.w. of ADM with 1 mM of phlorizin pretreatment (phlorizin + ADM) in T1R3-KO mice. (**B**) CT nerve responses of T1R3-KO mice to 500 mM of glucose (Glc), 500 mM of sucrose (Suc), 1 mM of SC45647 (SC), and 10 mM of NaCl 10–30 min after administration of vehicle (black bars) or 10^−6^ mol/kg b.w. of ADM without (red bars) or with (blue bars) pretreatment with 10^−7^ mol/kg b.w. of AM (22–52) (*n* = 7~10). (**C**) After pretreatment with SGLT1 inhibitor phlorizin (1 mM, lingual application), CT nerve responses of T1R3-KO mice to Glc, Suc, SC, and 10 mM of NaCl 10–30 min after administration of vehicle (black bars) or 10^−6^ mol/kg b.w. of ADM (red bars) (*n* = 7~8). SUM shows the sum of the magnitude of CT nerve responses to each stimulus and to 10 mM of NaCl. MIX indicates the magnitude of mixture of each stimulus with 10 mM of NaCl. Asterisks show significant differences (* *p* < 0.05, *** *p* < 0.001; Turkey HSD test following one-way ANOVA (**B**)). Under pretreatment with phlorizin, significant differences were not shown between vehicle controls and ADM administration (**C**). All data are presented as means ± SEMs.

**Figure 6 nutrients-15-02941-f006:**
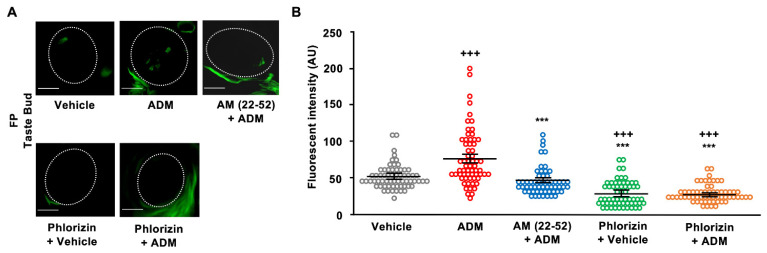
Adrenomedullin (ADM) enhanced 2-(N-(7-nitrobenz-2-oxa-1,3-diazol-4-yl)amino)-2-deoxyglucose (2-NBDG) uptake by taste cells in the intact tongue. (**A**) Representative images of FP taste buds after apical treatment with 10 mM 2-NBDG + 10 mM NaCl under different conditions (vehicle, ADM plus vehicle pretreatment (ADM), ADM plus AM (22–52) pretreatment (AM (22–52) + ADM), vehicle with phlorizin pretreatment (Phlorizin + Vehicle), ADM with phlorizin pretreatment (Phlorizin + ADM)). Dotted lines outline taste buds. Scale bars, 10 µm. (**B**) 2-NBDG uptake by FP taste cells. Each data point shows fluorescence intensity (arbitrary unit: AU) of a taste bud after apical treatment with 10 mM 2-NBDG + 10 mM NaCl under different conditions shown in (**A**) (*n* = 3, with 20~30 taste buds were analyzed in each animal). Symbols show significant differences between vehicle controls and ADM administration (^+++^ vs. vehicle, *** vs. ADM, *p* < 0.001, Tukey HSD test following one-way ANOVA). Summarized data are presented as means ± SEMs.

**Figure 7 nutrients-15-02941-f007:**
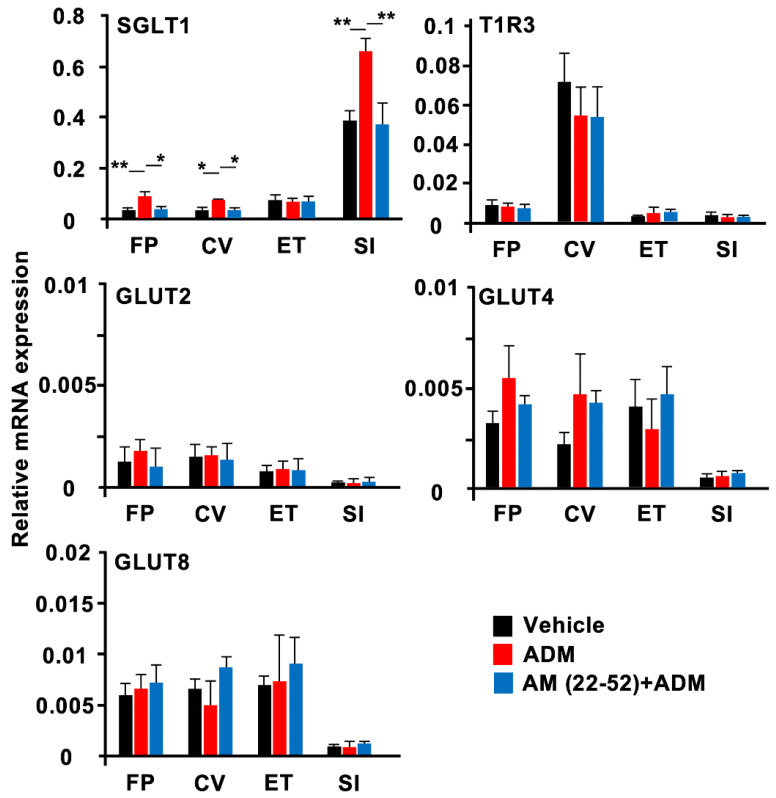
ADM increased relative mRNA expression of SGLT1 in mouse taste buds. Relative mRNA expressions of glucose transporters and T1R3 were shown under different conditions (vehicle (black bars) or 10^−6^ mol/kg b.w. of ADM without (red bars) or with (blue bars) pretreatment with 10^−7^ mol/kg b.w. of AM (22–52)) in each tissue (fungiform taste bud (FP), circumvallate taste bud (CV), epithelial tissue (ET), small intestine (SI)) (*n* = 7~12). Asterisks show significant differences (* *p* < 0.05, ** *p* < 0.01; Turkey HSD test following one-way ANOVA). All data are presented as means ± SEMs.

**Table 1 nutrients-15-02941-t001:** Nucleotide sequences for the primers used in RT-qPCR.

	Forward Primer	Reverse Primer
GAPDH	TGTGTCCGTCGTGGATCTGA	TTGCTGTTGAAGTCGCAGGAG
SGLT1	TCTGTAGTGGCAAGGGGAAG	ACAGGGCTTCTGTGTCTTGG
T1R3	CAAGGCCTGCAGTGCACAA	AGGCCTTAGGTGGGCATAATAGGA
GLUT2	GAGTTCCTTCCAGTTCGGCTATG	GTTCCACTGGATGACCGG
GLUT4	CTGTAACTTCATTGTCGGCATGG	AGGCAGCTGAGATCTGGTCAAAC
GLUT8	TCTGCATGTCAAGGGTGTGG	AGGGACAACGGTCAGTGTGAATAG

**Table 2 nutrients-15-02941-t002:** Co-expression rates of RAMP2, CRLR, and taste cell markers in FP (Figure 1).

RAMP2/T1R3	68.6% (70/102, *n* = 26)	T1R3/RAMP2	87.5% (70/80, *n* = 26)
CRLR/T1R3	78.4% (80/102, *n* = 26)	T1R3/CRLR	56.3% (80/142, *n* = 26)
ADM receptor /T1R3	67.6% (69/102, *n* = 26)	T1R3 /ADM receptor	89.6% (69/77, *n* = 26)
RAMP2/CA4	13.0% (6/46, *n* = 29)	CA4/RAMP2	11.8% (6/51, *n* = 29)
CRLR/CA4	76.1% (35/46, *n* = 29)	CA4/CRLR	36.1% (35/97, *n* = 29)
ADM receptor /CA4	13.0% (6/46, *n* = 29)	CA4 /ADM receptor	12.2% (6/49, *n* = 29)

**Table 3 nutrients-15-02941-t003:** Results of Student’s *t*-test for the effects of ADM or AM (22–52) on CT nerve responses (Figure 2B, Figure 3B and Figure 4A,B).

	Figure 2B	Figure 3B	Figure 4A	Figure 4B
Glc	*t*(12) = 5.37 *p* < 0.001	*t*(12) = −0.507 *p* = 0.311	*t*(23) = −4.84 *p* < 0.001	*t*(18) = 0.951 *p* = 0.177
Suc	*t*(12) = 2.52 *p* = 0.013	*t*(12) = −1.37 *p* = 0.099	*t*(12) = −3.21 *p* = 0.003	*t*(12) = −1.467 *p* = 0.168
SC	*t*(12) = 0.875 *p* = 0.199	*t*(12) = −0.558 *p* = 0.293	*t*(15) = 0.095 *p* = 0.463	*t*(15) = −0.488 *p* = 0.316
NaCl	*t*(12) = 0.097 *p* = 0.462	*t*(12) = −0.811 *p* = 0.217	*t*(19) = 0.034 *p* = 0.487	*t*(15) = 1.58 *p* = 0.068
KCl	*t*(12) = −0.951 *p* = 0.18	*t*(12) = 0.37 *p* = 0.359	*t*(20) = −0.794 *p* = 0.218	*t*(15) = −0.074 *p* = 0.471
HCl	*t*(12) = −1.88 *p* = 0.084	*t*(12) = −1.16 *p* = 0.133	*t*(16) = −0.866 *p* = 0.2	*t*(13) = −0.209 *p* = 0.419
QHCl	*t*(12) = −0.051 *p* = 0.48	*t*(12) = 1.04 *p* = 0.16	*t*(20) = 0.244 *p* = 0.405	*t*(15) = 0.356 *p* = 0.363
MPG	*t*(12) = 0.591 *p* = 0.283	*t*(12) = −0.147 *p* = 0.443	*t*(18) = −0.075 *p* = 0.471	*t*(14) = −0.285 *p* = 0.39

**Table 4 nutrients-15-02941-t004:** Results of two-way ANOVA test for the effect of ADM on CT nerve responses in B6 mice (Figure 2C).

	Treatment	Concentration	Interaction
Glc	*F*_(1,72)_ = 37.9 *p* < 0.001	*F*_(5,72)_ = 202 *p* < 0.001	*F*_(5,72)_ = 3.13 *p* < 0.05
NaCl	*F*_(1,60)_ = 2.14 *p* = 0.149	*F*_(4,60)_ = 273 *p* < 0.001	*F*_(4,60)_ = 0.625 *p* = 0.647

**Table 5 nutrients-15-02941-t005:** Results of two-way ANOVA test for time-dependent changes in CT nerve responses after ADM treatment (Figure 2D).

	Treatment	Time	Interaction
Glc	*F*_(1,84)_ = 29.9 *p* < 0.001	*F*_(6,84)_ = 8.15 *p* < 0.001	*F*_(6,84)_ = 5.78 *p* < 0.001
NaCl	*F*_(1,84)_ = 0.032 *p* = 0.755	*F*_(6,84)_ = 0.709 *p* = 0.961	*F*_(6,84)_ = 0.136 *p* = 0.842

**Table 6 nutrients-15-02941-t006:** Results of two-way ANOVA test for the effect of AM (22–52) on CT nerve responses in B6 mice (Figure 3C).

	Treatment	Concentration	Interaction
Glc	*F*_(1,72)_ = 0.751 *p* = 0.389	*F*_(5,72)_ = 131 *p* < 0.001	*F*_(5,72)_ = 0.54 *p* = 0.745
NaCl	*F*_(1,60)_ = 0.181 *p* = 0.672	*F*_(4,60)_ = 340 *p* < 0.001	*F*_(4,60)_ = 0.376 *p* = 0.825

**Table 7 nutrients-15-02941-t007:** Results of two-way ANOVA test for time-dependent changes in CT nerve responses after AM (22–52) treatment (Figure 3D).

	Treatment	Time	Interaction
Glc	*F*_(1,84)_ = 0.032 *p* = 0.858	*F*_(6,84)_ = 0.709 *p* = 0.643	*F*_(6,84)_ = 0.136 *p* = 2.21
NaCl	*F*_(1,84)_ = 2.36 *p* = 0.128	*F*_(6,84)_ = 0.113 *p* = 0.995	*F*_(6,84)_ = 0.253 *p* = 0.957

**Table 8 nutrients-15-02941-t008:** Results of one-way ANOVA test for the effect of ADM and AM (22–52) on mRNA expression (Figure 7).

	FP	CV	NT	SI
SGLT1	*F*_(2,29)_ = 6.134 *p* = 0.006	*F*_(2,29)_ = 5.525 *p* = 0.009	*F*_(2,30)_ = 0.029 *p* = 0.972	*F*_(2,26)_ = 8.997 *p* = 0.001
T1R3	*F*_(2,24)_ = 0.501 *p* = 0.612	*F*_(2,27)_ = 0.406 *p* = 0.670	*F*_(2,25)_ = 0.291 *p* = 0.750	*F*_(2,20)_ = 0.233 *p* = 0.794
GLUT2	*F*_(2,23)_ = 0.322 *p* = 0.728	*F*_(2,26)_ = 0.031 *p* = 0.970	*F*_(2,26)_ = 0.016 *p* = 0.984	*F*_(2,26)_ = 0.099 *p* = 0.906
GLUT4	*F*_(2,18)_ = 1.019 *p* = 0.381	*F*_(2,19)_ = 1.361 *p* = 0.280	*F*_(2,26)_ = 0.412 *p* = 0.667	*F*_(2,27)_ = 0.340 *p* = 0.715
GLUT8	*F*_(2,17)_ = 0.171 *p* = 0.845	*F*_(2,16)_ = 1.638 *p* = 0.225	*F*_(2,26)_ = 0.248 *p* = 0.782	*F*_(2,32)_ = 0.248 *p* = 0.782

## Data Availability

The data that support the findings of this study are available from the corresponding authors, S.I., R.Y., and Y.N., upon reasonable request.

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
