# Peer review of "Adrenomedullin Enhances Mouse Gustatory Nerve Responses to Sugars via T1R-Independent Sweet Taste Pathway"

_nutrients, 2023, doi:10.3390/nu15132941_

Round 1

Reviewer 1 Report

This paper Adrenomedullin enhances mouse gustatory nerve responses to sugars via T1Rs-independent sweet taste pathway examined the effect of Adrenomedullin (ADM), a multifunctional regulatory peptide, on sugar sensing in mice. The article demonstrates a contribution to the field and is in line with readers' interests of nutrients. However, there are still some shortcomings that need to be further improved or explained.

Comments:

Q1. In the abstract section, the expressions of since it affects expression of sodium-glucose cotransporter (SGLT1) in the rat enterocytes seemed  to have no obvious connection with the context before and after.

Q2. The research significance and innovation are suggested to be supplemented in the last paragraph of introduction.

Q3. Line 113, Why are there two ethics approval numbers?

Q4. Line 115, Why were only male animals selected.

Q5. Line 229, Why are different methods used in the analysis of data significance between groups? What are the advantages?

Q6. Figure 2C, is the Suc group with different Suc concentrations lost?

Q7. Why are the following tables comparing Glc and NaCl groups? Did the Suc group present more expressions?

Q8. Why was Western blotting not employed to detect the expressions of specific proteins? The corresponding protein expression maps might improve the reliability of these results.

Author Response

Dear Reviewer 1

Thank you and the reviewers for your careful consideration of our manuscript titled “Adrenomedullin enhances mouse gustatory nerve responses to sugars via T1R-independent sweet taste pathway.” for publication in Nutrients special issue "The Importance of Taste on Dietary Choice: Modulation of Taste Sensitivity”. Following the helpful comments of the reviewers, we have revised our manuscript and had it reviewed by an English-language editor. We hope that you and reviewers will consider this revised version suitable for publication in this Nutrients special issue.

Q1. In the “abstract” section, the expressions of “since it affects expression of sodium-glucose cotransporter (SGLT1) in the rat enterocytes” seemed to have no obvious connection with the context before and after.

  1. We have revised the “abstract” as shown below.

Original Manuscript (Lines 18 - 20)

“On the tongue, the T1Rs-independent (Glucose transporters, KATP Channel) pathway detects only sugars whereas the T1Rs-dependent (T1R2/T1R3) pathway can broadly sense various sweeteners.”

Revised Manuscript (Lines 18 - 20)

“On the tongue, the T1R-independent pathway [comprising glucose transporters, including sodium-glucose cotransporter (SGLT1), and KATP channel] detects only sugars, whereas the T1R-dependent (T1R2/T1R3) pathway can broadly sense various sweeteners.”

Original Manuscript (Lines 24 - 26)

“Here, we examined the effect of Adrenomedullin (ADM), a multifunctional regulatory peptide, on sugar sensing in mice since it affects expression of sodium-glucose cotransporter (SGLT1) in the rat enterocytes.”

Revised Manuscript (Lines 24 - 26)

“Here, we examined the effect of adrenomedullin (ADM), a multifunctional regulatory peptide, on sugar sensing in mice since it affects expression of SGLT1 in the rat enterocytes.”

Q2. The research significance and innovation are suggested to be supplemented in the last paragraph of introduction.

A. I appreciate your suggestion. I have added the following sentences to line 144:

“To our knowledge, no reports thus far have focused on the differences in function between the T1R-dependent and -independent pathways. This is the first study to investigate the regulating factor that selectively modulates newly found sweet taste pathways via SGLT1 in mouse taste buds. The elucidation of functional differences between these sweet taste pathways will shed new light on the development of anti-obesity treatment.”

Q3. Line 113, Why are there two ethics approval numbers?

A. One (A19-179-1) was approved for “Research and Development Center for Five-Sense Devices in Kyushu University”, and another (A21-360-2) was approved for “Section of Oral Neuroscience Graduate School of Dental Science in Kyushu University”. All experiments were done in both affiliations. We have clarified this in the manuscript.

Q4. Line 115, Why were only male animals selected.

A. We used male mice to avoid effects of sex differences. Especially, ADM’s gene is reported to be directly activated by estrogen receptor in mice1. I have added this information to lines 156-157.

  1. Watanabe, H. et al. The estrogen-responsive adrenomedullin and receptor-modifying protein 3 gene identified by DNA microarray analysis are directly regulated by estrogen receptor. J. Mol. Endocrinol. 36, 81–89 (2006).

Q5. Line 229, Why are different methods used in the analysis of data significance between groups? What are the advantages?

A. We have previously reported several studies on mechanisms of taste transduction using CT nerve recordings2, 3, 4, 5, 6, 7 and more, 2-NBDG uptake7, and qRT-PCR8, and this work is comparable. We thus used statistical analysis appropriate for each data type, as noted in reports cited above.

  1. Kawai, K. et al. Leptin as a modulator of sweet taste sensitivities in mice. Proc. Natl. Acad. Sci. U. S. A. 97, 11044–11049 (2000).
  2. Allais, L. et al. Transient receptor potential channels in intestinal inflammation: What is the impact of cigarette smoking? Pathobiology vol. 84 1–15 (2016).
  3. Niki, M. et al. Modulation of sweet taste sensitivities by endogenous leptin and endocannabinoids in mice. J. Physiol. 593, 2527–2545 (2015).
  4. Shigemura, N. et al. Angiotensin II modulates salty and sweet taste sensitivities. J. Neurosci. 33, (2013).
  5. Sukumarana, S. K. et al. Taste cell-expressed α-glucosidase enzymes contribute to gustatory responses to disaccharides. Proc. Natl. Acad. Sci. U. S. A. 113, 6035–6040 (2016).
  6. Yasumatsu, K. et al. Sodium-glucose cotransporter 1 as a sugar taste sensor in mouse tongue. Acta Physiol. (2020).
  7. Takai, S. et al. Effects of insulin signaling on mouse taste cell proliferation. PLoS One. 14: e0225190 (2019).

Q6. Figure 2C, is the Suc group with different Suc concentrations lost?

A. We did not measure Suc data because the dynamics of changes of responses to Suc are almost the same as to Glc, as noted in our previous studies6,7. Suc can activate sweet taste cells via T1R2/T1R3 directly and glucose transporters, including SGLT1, indirectly. In this study, ADM administration did not increase CT nerve responses to artificial sweetener SC45647, indicating that ADM had no effects on the T1R2/T1R3 pathway. This indicates that ADM enhances responses to sugars via glucose transporters. We also found that sweet taste cells expressed disaccharide-digesting enzyme α-glucosidase, which cleaves Suc to Glc and fructose. Because only Glc generated from Suc by α-glucosidase could serve as substrate for T1R-independent sugar-sensing pathways6, we omitted Suc data.

Q7. Why are the following tables comparing Glc and NaCl groups? Did the Suc group present more expressions?

A. Please see our response to Q6.

Q8. Why was Western blotting not employed to detect the expressions of specific proteins? The corresponding protein expression maps might improve the reliability of these results.

A. I agree Western blotting methods are very suitable for investigating ADM effects for expression levels of SGLT1 and other molecules. However, this technique is very difficult to apply here because of the very small amount of specimen we can collect from mouse fungiform papillae (FP) and circumvallate papilla (CV). Therefore, researchers in my area of expertise do not use Western blotting methods. In this report, as alternatives, we examined uptake of 2-NBDG and mRNA expressions of T1R3 and glucose transporters.

Sincerely yours,

Shusuke Iwata

1851-1 Hozumi, Mizuho, Gifu 501-0296 Japan

Dept. Oral Physiol., Asahi Univ. Sch. Dent.

Shusuke Iwata, D.D.S., Ph.D.

E-mail: siwata@dent.asahi-u.ac.jp

Tel / Fax: +81-58-329-1412

Reviewer 2 Report

Iwata et al reported that adrenomedullin enhance T1R-independent glucoses-induced sweet response on the tongue.

Their finding about the new taste-modifier hormone is novel. It is, however, additional experiments are required to strengthen their conclusion.

[1] In Figure 1, why did they focus only on the fungiform papillae? They should check circumvallate papilla if there is no special reason.

[2] In Figure 1, why did they check SGLT1 expression as well. This information is quite important and well-related to Figs. 5-7.

[3] In Figure 3A, it is quite difficult to see fluorescent signals in the cells. They should provide better images.  

Author Response

Dear Reviewer,

Thank you and the reviewers for your careful consideration of our manuscript titled “Adrenomedullin enhances mouse gustatory nerve responses to sugars via T1R-independent sweet taste pathway.” for publication in Nutrients special issue "The Importance of Taste on Dietary Choice: Modulation of Taste Sensitivity”. Following the helpful comments of the reviewers, we have revised our manuscript and had it reviewed by an English-language editor. We hope that you and reviewers will consider this revised version suitable for publication in this Nutrients special issue.

[1] In Figure 1, why did they focus only on the fungiform papillae? They should check circumvallate papilla if there is no special reason.

A. In this study, we examined the effects of ADM on responses of the chorda tympani (CT) nerve, which innervates fungiform papillae. Because the circumvallate papilla is innervated by the glossopharyngeal nerve, we mainly examined only fungiform papillae in Figure 1 and Figure 3.

[2] In Figure 1, why did[n’t] they check SGLT1 expression as well. This information is quite important and well-related to Figs. 5-7.

A. I appreciate for your sound advice. Unfortunately, the species of primary antibody host of SGLT11 and CRLR are same (rabbit), so we could check only CRLR. We used T1R3-GFP mice as an alternative for glucose response in Figure 1.

  1. Yee, K. et al. Glucose transporters and ATP-gated K+ (KATP) metabolic sensors are present in type 1 taste receptor 3 (T1r3)-expressing taste cells. Proc. Natl. Acad. Sci. U. S. A. 108, 5431–5436 (2011).

[3] In Figure 3A, it is quite difficult to see fluorescent signals in the cells. They should provide better images.

A. I would like to answer about Figure 6A as well, because I am afraid this question may be asked about it also. I agree the resolution of each of the figures in this manuscript was low. I have now changed all figures to higher resolution.

Sincerely yours,

Shusuke Iwata

1851-1 Hozumi, Mizuho, Gifu 501-0296 Japan

Dept. Oral Physiol., Asahi Univ. Sch. Dent.

Shusuke Iwata, D.D.S., Ph.D.

E-mail: siwata@dent.asahi-u.ac.jp

Tel / Fax: +81-58-329-1412

Round 2

Reviewer 1 Report

Thanks for these modifications or explanations, I have no any other comments.